# Machine Learning Reveals Microbial Taxa Associated with a Swim across the Pacific Ocean

**DOI:** 10.3390/biomedicines12102309

**Published:** 2024-10-11

**Authors:** Garry Lewis, Sebastian Reczek, Osayenmwen Omozusi, Taylor Hogue, Marc D. Cook, Jarrad Hampton-Marcell

**Affiliations:** 1Department of Biological Sciences, University of Illinois at Chicago, Chicago, IL 60607, USA; glewis9@uic.edu (G.L.); srecze2@uic.edu (S.R.); 2Department of Biomedical Engineering, University of Illinois at Chicago, Chicago, IL 60607, USA; oomoz2@uic.edu; 3Department of Kinesiology, North Carolina Agriculture and Technical State University, Greensboro, NC 27411, USA; thogue@aggies.ncat.edu (T.H.); mdcook@ncat.edu (M.D.C.); 4Center of Integrative Health Disparities and Equity Research, North Carolina Agriculture and Technical State University, Greensboro, NC 27411, USA

**Keywords:** gut microbiome, host–microbe, machine learning, physical fitness

## Abstract

**Purpose:** This study aimed to characterize the association between microbial dynamics and excessive exercise. **Methods:** Swabbed fecal samples, body composition (percent body fat), and swimming logs were collected (n = 94) from a single individual over 107 days as he swam across the Pacific Ocean. The V4 region of the 16S rRNA gene was sequenced, generating 6.2 million amplicon sequence variants. Multivariate analysis was used to analyze the microbial community structure, and machine learning (random forest) was used to model the microbial dynamics over time using R statistical programming. **Results:** Our findings show a significant reduction in percent fat mass (Pearson; *p* < 0.01, R = −0.89) and daily swim distance (Spearman; *p* < 0.01, R = −0.30). Furthermore, the microbial community structure became increasingly similar over time (PERMANOVA; *p* < 0.01, R = −0.27). Decision-based modeling (random forest) revealed the genera *Alistipes*, *Anaerostipes*, *Bifidobacterium*, *Butyricimonas*, *Lachnospira*, *Lachnobacterium*, and *Ruminococcus* as important microbial biomarkers of excessive exercise for explaining variations observed throughout the swim (OOB; R = 0.893). **Conclusions:** We show that microbial community structure and composition accurately classify outcomes of excessive exercise in relation to body composition, blood pressure, and daily swim distance. More importantly, microbial dynamics reveal the microbial taxa significantly associated with increased exercise volume, highlighting specific microbes responsive to excessive swimming.

## 1. Introduction

Exercise is renowned for its positive impact on physical fitness, physiological well-being, and athletic performance, evidenced by improved body composition, reduced blood pressure, and enhanced performance metrics [1,2,3]. It has also been shown to enhance glucose uptake by upregulating glucose transporters in skeletal muscle and promoting glycogen storage, thus improving energy utilization and exercise performance [4]. While these benefits are well-documented, exercise also introduces physiological stress that may contribute to overtraining syndrome and cardiovascular risks, emphasizing the need for a holistic understanding of exercise [5,6]. The gut microbiome (GM), hosting a vast array of nearly 30 trillion microbial cells, emerges as a critical factor in this context [7]. Its functional capacity to regulate glucose homeostasis and influence metabolic pathways suggests a profound connection with the body’s response to exercise [8,9]. The relationship between exercise—its frequency and volume—and the GM is associated with body composition and cardiorespiratory fitness, suggesting a similar or shared pathway for muscular endurance and metabolic health [10,11].

Concurrently, exercise profoundly influences the GM, as evidenced by changes in alpha and beta diversity in human and murine models [12]. Adaptive GM changes feature increased diversity, with abundant short-chain fatty acid (SCFA)-producing bacterial orders like *Clostridiales* and taxa such as *Roseburia* [13,14]. These microbial populations contribute to metabolic homeostasis by producing SCFAs and enhancing immune regulation [15,16,17] and endurance exercise performance [18]. Improved endurance and strength may result from this synergy, modulating oxidative stress and inflammatory responses, enhancing metabolism, and optimizing energy expenditure during intense exercise [19]. Further underlining the GM’s importance, research in murine models reveals that a depleted GM can lead to decreased muscular endurance and glycogen availability, which are reversed with GM restoration [20]. Moreover, increased cardiorespiratory fitness (i.e., VO_2max_) has been linked to increased GM diversity and elevated butyrate levels [21]. Despite these positive associations, there is a critical need to explore how these effects might vary or reverse under conditions of excessive exercise (EE). According to O’Keefe [22], EE can be induced with as little as four to five hours of vigorous exercise within a single week.

While exercise is widely recognized for its positive health benefits, there is a threshold beyond which exercise can become detrimental. Individuals who engage in EE have a higher incidence of illness and overuse injuries, with the most common injury being stress (fatigue) bone fractures. Furthermore, EE is associated with poor nutritional and psychological health, decreased energy availability, and worse exercise performance [23]. At minimum, sufficient recovery from EE involves at least one full rest day per week [22]. Assessing EE in relation to the GM, Yuan et al. [24] observed that EE is associated with decreased microbial diversity when studied within a murine model. Importantly, EE affects the GM by increasing inflammation and gut permeability, thereby increasing the risk of pathogen entry into the body [25]. 

To further understand the impact of EE and the GM, we analyzed a case study of a 51-year-old man who attempted to traverse the Pacific Ocean, logging 1212.59 nautical miles over the course of 107 days. Our study focuses on a continuous time-series analysis, providing concise change points for composition, structure, performance, and physiology. Based on the assumption that certain microbial profiles are optimized for endurance performance, we hypothesize that structural and compositional changes in the GM, driven by the high frequency and volume of swimming, can predict the overall time course of exercise. To our knowledge, this is the first study to monitor microbial dynamics and EE at this magnitude. Our research provides an interpretation of the GM when exercise transitions from beneficial to adverse, which is often overlooked when studying host–microbe interactions and exercise.

## 2. Methods

### 2.1. Study Design

A 51-year-old Caucasian man embarked on a 107-day swim across the Pacific Ocean, encompassing 1212.59 nautical miles. The participant received an explanation of the study and signed a written informed consent form that had been approved by the central institutional review board within the Department of Energy (protocol DOE000277). The study participant swam 8 h on average, rested on a boat, and returned to the same area the following morning to resume swimming. A physician on board monitored several health markers daily, including body fat composition (waist-to-height ratio), arterial oxygen saturation via a pulse oximeter (SaO_2_), heart rate (HR), and systolic and diastolic blood pressure, to assess changes in both physiological and physical measures. Over the 107-day swim, swabbed fecal samples were collected at 94 timepoints (n = 94). GPS technology was used to record daily swimming distances and latitudinal and longitudinal coordinates; wind speed and direction, wave height and direction, and weather conditions were also measured. 

### 2.2. Sample Processing and Sequencing

Daily swabbed fecal samples were collected from the participant. Fecal samples were stored in a −20 °C freezer during the swim. Following the conclusion, samples were shipped on dry ice to Argonne National Laboratory and stored in a −80 °C freezer until downstream processing. Genomic DNA was extracted following Qiagen’s QIAamp Fast DNA Stool Mini Kit (Germantown, MD, USA) suggested protocol. Libraries were generated using forwarded barcoded primers according to the American Gut Project, and the Greengenes database version 13.8 was used for 16S rRNA sequencing. Utilizing the V4 region of the 16S rRNA gene, microbial sequence data were quality-filtered and assigned to corresponding samples based on their 12-bp error-correcting Golay barcodes using QIIME2. Amplicon sequence variants were generated using deblur, and the amplicon sequence variants (ASVs) were analyzed using Quantitative Insight into Microbial Ecology 2 (QIIME2) and R Statistical Software (v4.2.1). Microbial sequences appearing only once were discarded and not included in downstream analyses.

### 2.3. Statistical Analysis

R statistical language (v4.2.1) and RStudio (v4.2.1) were used for all analyses. The participant’s swim was mapped with coordinate data using the R packages ‘sf’, ‘rnaturalearth’, ‘googleway’, ‘cowplot’, ‘ggrepel’, and ‘ggspatial’. Pearson correlations of swim (daily distance), blood pressure, and body fat composition were plotted over time with a statistical significance cutoff of *p* < 0.05. Alpha diversity measures (inverse Simpson and Shannon indices) were generated using the R package ‘microbiome’, and linear regression models (LRMs) were applied to test whether changes over time were significant (*p* < 0.05). Wilcoxon tests were performed to test if differences between means of groups were statistically significant (*p* < 0.05).

Bray–Curtis distances were generated to assess dissimilarities in microbial community structure using the R packages ‘microbiome’, ‘phyloseq’, and ‘vegan’. K-means clustering, a type of unsupervised learning model, was applied to microbial composition to identify microbial subcommunity prevalence over time. This method groups similar observations by placing data points into groups (k’s) that are most similar based on average values. K-means clustering was applied to generate 1–10 microbial subcommunities using Bray–Curtis distances. To determine the best number of groups (k), we used a silhouette analysis, which compares how closely an observation matches its group versus others. This analysis, conducted with the ‘fpc’ and ‘cluster’ R packages, assigns scores from −1 to 1 to each observation, with a high score indicating a good fit within its group. Clusters were visualized with a dimensional reduction using principal coordinate analysis (PCoA), and a permutational analysis of variance (PERMANOVA) test assessed if community-level differences were significant. Additionally, a density plot over time determined temporal and dominance interactions between subcommunities. 

Decision trees (supervised learning) were applied to identify microbial taxa important in shaping microbial community composition and structure over time using the R package ‘randomForest’. Random forest (RF) is a machine learning classifier that implements a series of interconnected and hierarchical decisions to analyze data. Multiple decision trees (i.e., RFs) are generated by building upon a consensus model that subsequently quantifies a variable’s importance in the model’s accuracy. The RF models used out-of-bag (OOB) estimates for the validation and accuracy assessment. Taxa were ranked based on their feature scores, calculating the mean decrease in the model’s accuracy when a given taxon was removed from training and the subsequent effect on the model’s performance. Furthermore, Spearman rank correlations were generated to observe whether taxa identified by RF models changed significantly (*p* < 0.05) over time. All figures were generated using the R package ‘ggplot2’ and related packages ‘ggsci’, ‘ggpubr’, ‘ggpmisc’, and ‘patchwork’. General data restructuring was performed in tandem with the ‘dplyr’ and ‘broom’ packages.

## 3. Results

### 3.1. Excessive Exercise Significantly Alters Microbial Community Structure

Over the course of 107 days, the participant swam a cumulative distance of 1212.59 nautical miles (N), averaging 17.32 N/day (±8.44) (Figure 1A). Of the 107 days at sea, the participant spent 70 days actively swimming and 37 days resting, engaging in continuous exercise for an average of 6.36 days and ranging from a minimum of 2 days to a maximum of 14 days. It was suspected that the participant would be unable to keep up with the demands of the exercise over such a prolonged duration, so a linear regression was used to analyze changes in the participant’s swim over time. The participant showed a significant reduction (Pearson, *p* < 0.05, R = −0.48) in daily swim distance over time, dropping below his overall average after day 57 (Figure 1B). Furthermore, when split into quartiles to assess the participant’s change in performance from start to finish, each quartile representing approximately 27 days, we observed an average swim distance of 22.39 N/day (±11.78) and 12.09 N/day (±5.25) within the first and last quartiles, respectively (Table 1, Appendix A), demonstrating the participant swam nearly twice as far at the beginning of the swim than at the end.

To measure exercise activity, heart rate and blood pressure were measured and assessed using a linear regression to observe changes over time. On average, the participant’s heart rate was 65.67 bpm (±6.64) (Appendix A); resting blood pressure was 128.28 mmHg (±11.50) (Figure 1C), and body fat composition was 19.31% (±1.41) (Figure 1D). While the participant’s average resting heart rate and body fat composition were within normal ranges [26,27], the average blood pressure was elevated [28]. There was a significant (Pearson, *p* < 0.05, R = 0.44) change in the participant’s blood pressure over time (Figure 1C). Comparing the participant’s systolic blood pressure between the start and end, we observed an increase from a healthy 115.40 mmHg (±5.27) in the first quartile to a stage 1 hypertensive 135.86 mmHg (±15.70) in the fourth. A linear regression of body fat composition also showed a significant (Pearson, *p* < 0.05, R = −0.85) decrease over time (Figure 1D). Interestingly, the heart rate (Appendix A) did not change significantly (Pearson, *p* > 0.05). The participant’s arterial oxygen saturation (SaO_2_) primarily remained at an expected level of 97–98% when measured, exhibiting no significant (Pearson, *p* > 0.05) change over time. However, notable exceptions occurred on day 68 and day 90, with SaO_2_ dropping to 93%.

The GM community composition was assessed following the filtering of single ASVs, with 6,237,032 sequences identified for 94 samples with a mean ASV count of 7286.25, a minimum of 10, and a maximum of 442,567. Among the identified ASVs, the phyla Firmicutes (45.44%) and Bacteroidetes (40.95%) comprised approximately 86.39% of the microbial composition on average. Bacterial diversity—both alpha and beta diversity—was analyzed to characterize the broad microbial diversity and structure changes. Shannon (emphasizing microbial richness) and inverse Simpson (emphasizing microbial evenness) indices were analyzed to measure alpha diversity. Both alpha diversity measures were fitted against a linear regression; however, neither measure of alpha diversity showed a significant change over time (LRM, *p* > 0.05; Appendix A), suggesting microbial diversity was relatively stable during the participant’s swim.

The microbial community structure was assessed in association with time by fitting changes in Bray–Curtis dissimilarities across consecutive timepoints against a generalized linear regression. This approach allowed flexibility in dealing with the non-normal distribution of ecological data by accommodating the unique variance structures often present in microbiome studies. Over the swim, the microbial community became increasingly similar, marked by a 44.24% reduction in beta diversity from start to end (GLM, *p* < 0.01, R = −0.27; Appendix A). While alpha diversity remained stable, beta diversity altered significantly, indicating microbial restructuring and a different microbial subcommunity by the end of the swim.

### 3.2. Machine Learning Reveals Microbial Taxa That Shape Swim Time Course

To further understand the association between the participant’s swim and the GM, an RF model was employed. For the participant, an RF model was used to train and predict the swim time course based on microbial composition at each timepoint. Using an out-of-bag model, the microbial composition readily predicted the swim time course (OOB; R = 0.89), demonstrating that changes in the GM highly reflected the participant’s swim. Assessing importance features, the top 10 ASVs were identified and binned at the genus level. For the identified ASVs, the genera *Alistipes* (11.01%)*, Ruminococcus* (9.10%)*,* and *Bifidobacterium* (9.37%) accounted for 29.48% of the model’s accuracy (Figure 2A). Applying a linear regression to their abundances across time, *Ruminococcus* and *Bifidobacterium* exhibited significant changes in relative abundance, decreasing by 40.52% and 86.97%, respectively, while *Alistipes* significantly increased by 73.47% (Figure 2B). The bacterial order contributing the most to the model’s accuracy was *Clostridiales* (58.31%). These results imply that this order and the previously identified genera may significantly influence the physiological mechanisms underlying the participant’s swim performance. 

As beta diversity suggested a shift in the microbial community structure over time, microbial co-occurrences were further assessed using unsupervised machine learning. K-means clustering was employed to identify microbial subcommunities (i.e., enterotypes) as it can group data into distinct clusters based on similarities in the microbial community structure. A silhouette analysis identified the optimal number of K-means clusters based on Bray–Curtis distances and predicted the overall microbial variation within the clusters. When grouped into two clusters (k = 2), more than 70% of the microbial variation was explained (Appendix A). Patterns in the microbial abundance and taxa over time were also examined. A principal coordinate analysis (PCoA) demonstrated that the two enterotypes were significantly (PERMANOVA, *p* < 0.05, R = 0.74; Figure 3A) distinct from one another, suggesting two distinct microbial structures over the duration of the swim. Visualizing the distributions of the enterotypes over time revealed that the earlier half of the swim was dominated by enterotype 1, while the latter was dominated by enterotype 2 (Figure 3B). Examination of the subcommunities’ relative abundances revealed the dominance of Firmicutes and Bacteroidetes, which collectively comprised 77.71% of enterotype 1 (38.58% and 39.13%, respectively) and 81.11% of enterotype 2 (36.80% and 44.31%, respectively). Notably, the taxa *Prevotella*, *Roseburia*, and *Sutterella* were the primary discriminators between the two subcommunities (RF OOB, R = 0.90, Figure 4A), with a significant percent increase in the relative abundance of *Prevotella* (78.95%), *Roseburia* (44.08%), and *Sutterella* (44.68%) from enterotype 1 to enterotype 2 (Wilcoxon, *p* < 0.05, Figure 4B).

## 4. Discussion

This case study demonstrated that the GM is dynamic and responds to exercise of high volume and frequency. Notably, a shift between distinct microbial subcommunities was initially indicated by beta diversity and subsequently identified via K-means clustering. This microbial shift corresponded with a decline in key performance metrics, including swim distance, and was linked to changes in blood pressure and body fat composition. Additionally, GM composition accurately predicted the time course of exercise, potentially serving as a non-invasive biomarker for tracking endurance exercise performance. These findings highlight the influence of EE on GM composition. Our study emphasizes the value of examining the longitudinal relationship between exercise and the GM, revealing insights into microbiome dynamics that may not be apparent in comparative analyses.

### 4.1. Physiological Measures Indicate Maladaptive Responses to EE

We observed that the participant’s body fat percentage significantly decreased by approximately 3% during the first 60 days of the swim, consistent with reported levels of fat loss from high-volume exercise [29,30]. This may indicate that the participant was unable to maintain his energy balance, burning more calories than he was able to replace [31]. Notably, despite high exercise volume, the participant’s body fat percentage appeared to stabilize after day 60, potentially causing the body to burn fewer calories to preserve energy while increasing fatigue and risk of permanent organ damage, especially to the heart [32]. This raises important considerations regarding the participant’s cardiovascular health. The participant’s average heart rate remained at 65.67 bpm (±6.64) throughout the study. Ordinarily, one might anticipate a short-term increase in heart rate during active periods as an adaptation to supply the muscles with more blood [33] or a long-term decrease in resting heart rate as a marker of enhanced cardiovascular efficiency to exercise [34]. However, an unchanging heart rate over time suggests that the expected adaptations may not have materialized in response to EE.

Despite SaO_2_ also not significantly changing, the participant experienced clinically significant changes (≥4%), indicating hypoxemia where the blood was deprived of adequate oxygen supply [35]. Simultaneously, our analysis revealed that the participant’s blood pressure increased significantly from normotensive levels (~120 mmHg) to stage 2 hypertension (≥140 mmHg), peaking at 156 mmHg on multiple occasions and averaging 128.28 mmHg overall, which is indicative of EE (Figure 1C). This increase contrasts with the typical post-exercise blood pressure reduction of 2–4 mmHg in normotensive individuals and 5–8 mmHg in hypertensive individuals [28]. An increase in blood pressure may indicate a heightened effort to supply oxygen and nutrients to muscles during the participant’s intense, prolonged swimming sessions [36]. The blood pressure increase may be related to a lack of energy availability coinciding with a significant reduction in body fat, which indicates that caloric intake was insufficient to maintain fat mass [37]. Overall, this suggests that the participant’s heart was under considerable strain while under the demands of the exercise [28], with a potentially increased risk of developing cardiovascular disease. 

Concurrently, a decline in swim performance over time was observed (Figure 1B), suggesting fatigue and insufficient recovery, characteristics often associated with EE [23]. While we did not directly measure the participant’s exercise intensity, swimming typically has a METs (metabolic equivalents) score in the range of 6 to 11, considered to be vigorous intensity [38,39]. Based on this, it is likely that our participant engaged in intense exercise. However, without specific METs measurements for our participant, this inference should be made cautiously. The participant’s blood pressure readings, which frequently exceeded 150 mmHg, might reflect such physiological strain, but this cannot be conclusively established from our data. Together, the observed rise in blood pressure, alongside a significant decrease in body fat composition and diminished swimming performance, raises concerns about the sustainability of the participant’s exercise, hinting at an imbalance where the physical demands of exercise may surpass the body’s capacity to adapt and recover.

### 4.2. Microbial Diversity and EE

Previous studies have indicated that professional athletes, such as rugby players, exhibit a higher GM alpha diversity than non-athletes [13]. This enhanced diversity is often attributed to their distinct physical fitness levels and dietary habits [40]. Often, greater alpha diversity is associated with beneficial health outcomes such as improved bowel health [41] and reduced incidence of various acute and chronic diseases [42]. However, the impact of exercise on GM diversity is not uniform across all studies, and exercise studies have yielded mixed results. Some report increased alpha diversity following exercise, while more than half observe no significant changes [43]. This inconsistency underscores a notable gap in our understanding of how prolonged, high-volume exercise influences GM diversity, especially in professional and endurance athletes.

In the case of our 51-year-old participant, no change in alpha diversity was observed during the swim. This result coincides with a previous study of triathlon athletes, where no significant change was reported in fecal microbiota richness and evenness pre- and post-race [44]. This similarity suggests that certain types of high-volume exercise may not notably affect alpha diversity. However, an unchanging alpha diversity may align with the participant’s health status, which included a rise in blood pressure to hypertensive levels and potential signs of starvation [28,45]. However, this pattern is inconsistent with findings from Sun et al. [46], who reported a significant, inverse relationship between alpha diversity (Shannon index) and systolic blood pressure. Finally, considering the participant’s age, the literature suggests diminished responsiveness of the GM, particularly alpha diversity, to exercise in older individuals. Soltys et al. [47] noted that older adult athletes engaging in endurance exercise did not experience significant changes in their GM, indicating a potential age-related decline in GM responsiveness to exercise. This suggests that our participant’s age may have contributed to the observed stability in alpha diversity despite his high volume of physical activity.

Additionally, a marked reduction in beta diversity was observed in our participant, indicative of a shift toward a more similar and stable microbial community over the study’s time course. This aligns with Kern et al. [48], who report a significant reduction in beta diversity in obese participants following vigorous exercise over 6 months. A similar reduction in beta diversity was also observed in triathlon athletes [44]. Increased GM homogeneity may indicate dysbiosis, because a healthy microbiome, capable of responding to its environment, is characterized by both dynamic microbial turnover and compositional stability [49].

Our results implicate a dichotomous state of the GM post-exercise: resilient in maintaining species richness yet potentially trending toward a less diverse (more stable) composition. This reflects the complex interactions between the maintenance of microbial homeostasis and the onset of dysbiosis, as the GM has been observed to stabilize with beneficial or harmful bacteria [50]. Consequently, our findings invite further exploration into the functional consequences of exercise-induced shifts in the gut microbiota and their longitudinal impact on host well-being.

### 4.3. GM Alterations: Microbial Adaptations and Maladaptations in Response to EE

A significant increase was observed in *Alistipes* over time. While *Alistipes* has been reported to exert some beneficial effects, it is more commonly linked with unfavorable health conditions such as hypertension [51,52]. Importantly, we report a significant increase in the prevalence of *Alistipes* in our participant’s GM over time. This observed relationship prompts additional research on the possible contributory role of *Alistipes* in blood pressure regulation, especially under the stress of high-volume, prolonged exercise.

We also observed a significant decrease in beneficial SCFA-producing bacteria such as *Bifidobacterium* and *Ruminococcus* throughout the participant’s swim. The reduction in *Bifidobacterium*, known for its health-promoting effects including immunomodulation and maintaining gut-barrier integrity, may be related to our participant’s declining performance, as these bacteria contribute to overall health and energy metabolism, although they are typically enriched by exercise [53,54]. Similarly, *Ruminococcus*, involved in complex carbohydrate breakdown, is associated with many exercise models and often with beneficial health outcomes. Its response to exercise (i.e., change in relative abundance) may indicate alterations in carbohydrate metabolism and energy balance that impact individuals’ performance under physical stress [55,56]. Considering this, *Ruminococcus* is implicated as a microbial biomarker for physical performance in our study, just like *Bifidobacterium*, as these SCFA-producers significantly decreased alongside the participant’s daily swim distance.

Examination of the GM revealed two distinct subcommunities: the first subcommunity dominated the initial 57 days of the swim, with subcommunity two superseding it from day 58 onward. Further investigation into these microbial subcommunities revealed a significant increase in characteristic taxa such as *Prevotella*, *Roseburia*, and *Sutterella*. An increase in *Prevotella*, often implicated in health concerns like inflammation, hypertension, and dysbiosis [57], occurred alongside increased blood pressure in our participant, suggesting that *Prevotella* may act as a useful biomarker for hypertension. On the other hand, *Roseburia* is associated with gut health and lower disease rates [58,59], which may indicate a protective role in the GM of our participant under the physiological stress of EE. *Prevotella* and *Roseburia* are associated with high sodium intake, which raises the possibility that salt water could be a contributing factor to the GM [60]. The role of *Sutterella*, however, remains more elusive, being prevalent in the human gastrointestinal tract as a commensal organism with mild anti- and pro-inflammatory properties [61,62,63,64]. The prevalence of *Sutterella* significantly increased over time, which may indicate an adaptive immune or inflammatory response of the GM to the conditions faced by our participant.

Our case study reveals dynamic shifts in the participant’s GM, particularly in the latter phase of the swim, where there was a notable shift toward microbial profiles commonly linked to physiological stress [51,52,57]. Accompanying this microbial shift were significant changes in performance and physiological markers, suggesting a link between EE and microbial composition. This likely reflects a selection pressure or environmental influence that caused certain microbial species to become more prevalent, indicating potential microbial adaptation to the swimming conditions.

### 4.4. Limitations and Future Directions

This study is subject to several limitations. Primarily, the one-of-a-kind design of this study, involving swimming across the Pacific Ocean, is difficult to replicate, although ultramarathon studies, examining exercise in relation to the GM, may closely replicate our data. Despite this challenge, our study provides a clear direction for future research of exercise-associated GM dynamics by identifying specific taxa and physiological changes. Another concern is the potential impact of swallowing sea water on GM composition and structure [60,65]. Despite this limitation, the observed changes in GM diversity are more likely to be diet-related.

A major limitation of this study is the absence of dietary data, as diet is known to influence the GM in ways akin to exercise, altering the microbial community structure and composition [66]. Undigested food particles act as variable substrates for metabolism by microbiota located in the colon, affecting GM diversity and function as they use these metabolites [67]. Consequently, it is unclear whether the observed GM changes are solely attributable to exercise or if dietary alterations coexisting with the participant’s exercise regimen played a role. It is possible that the initial decrease in body fat composition could have been from a lack of calorie consumption, while the stabilization may have resulted from increasing caloric intake [37]. Nevertheless, we observe a significant exercise-associated relationship with the GM characterized by changes in the relative abundance of many different microbial taxa and a dramatic shift in GM community structure.

Additionally, the absence of early blood pressure readings and VO_2max_ data—key indicators of cardiovascular fitness and exertion [36]—complicates full understanding of the participant’s physiology. Standard exercise intensity measurements like VO_2max_ could not be measured due to the unique study environment. However, despite this limitation, we still observed significant changes in the participant’s physiology, fitness, and performance coinciding with EE. The participant’s blood pressure escalated from healthy levels to stage 2 hypertension, which suggests a significant and maladaptive physiological change. Moreover, a significant decrease in the participant’s swim distance over time implies heightened fatigue and performance decline [23]. Together, these observations, despite the absence of VO_2max_ data, support the findings of adverse effects of EE on performance, reinforcing the link between physical activity, environmental factors, and dietary influences on the GM.

While we observed a decline in body fat composition akin to patterns seen in extreme endurance activities like arctic trekking and ultramarathons [68], which suggests metabolic adaptation [69], the study’s design precluded direct measurement of the participant’s BMR or energy expenditure, further limiting our analysis. Furthermore, our findings on alpha diversity following EE contrast with Yuan et al. [24], although this discrepancy may arise from interspecies differences or differing study parameters. Other limitations include the lack of generalizability of our results considering the substantial inter- and intra-variability of the GM [50]. Gender-specific responses to exercise, particularly in the context of EE, could not be explored with a single male participant.

This study highlights the significant effects of long-term, high-volume exercise on the GM, emphasizing the need for further exploration into how athletes’ training and lifestyle influence the GM, particularly in endurance sports with in-season and out-of-season periods. It advocates for future research to adopt controlled designs, engage diverse participants, and employ thorough data collection. To build directly on the findings presented here, subsequent studies should include dietary analysis, assessment of environmental effects, and better detailed physiological evaluations, all aimed at eventually refining health and performance optimization strategies for athletes.

## Figures and Tables

**Figure 1 biomedicines-12-02309-f001:**
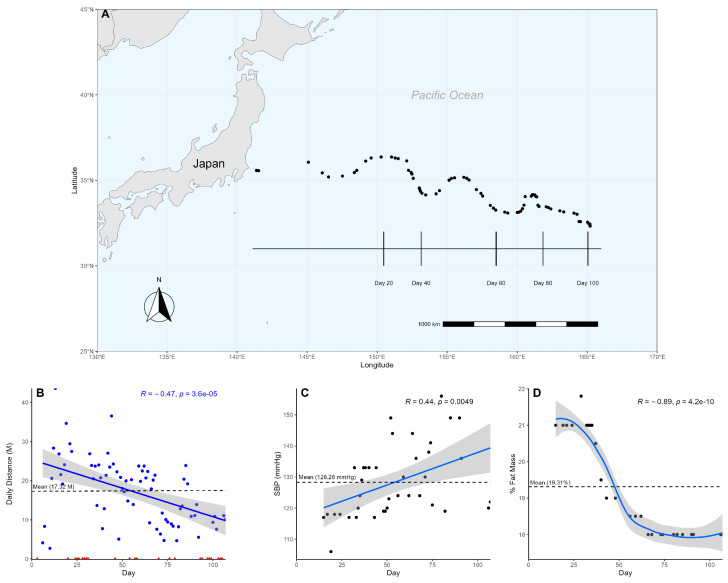
Participant’s swim across the Pacific Ocean. (**A**) Participant’s progress was tracked using longitudinal and latitudinal coordinates. Vertical lines mark the participant’s location at approximately 20-day intervals. Each dot represents a sample collection date. (**B**) The participant’s daily distance is graphed over time using a linear model. Non-swimming days are marked as red triangles, while swimming days are marked as blue dots. The non-swimming days were excluded from the model to ensure an accurate representation of the change in daily distance over time. (**C**) A linear model shows the participant’s systolic blood pressure over time. (**D**) The participant’s body fat composition, measured in percent fat mass, is graphed using a loess model over time.

**Figure 2 biomedicines-12-02309-f002:**
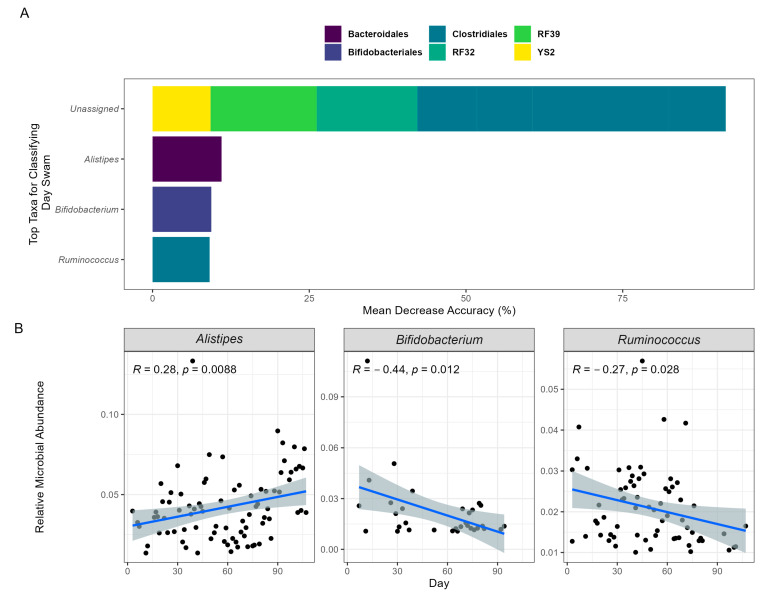
Machine learning predicts swim day based on GM composition over time. The prediction model employed RF and was validated using OOB of the GM composition data collected on 94 separate sample days. (**A**) The top 10 taxa that have the most influence on predicting model accuracy are presented. The taxa’s orders are categorized by color, and their genera are labeled on the *x*-axis. The “Unassigned” category includes bacteria not yet identified at the genus level. (**B**) The relative abundances of the identifiable top genera are depicted in a Spearman rank correlation over time via a linear regression shaded with 95% confidence intervals.

**Figure 3 biomedicines-12-02309-f003:**
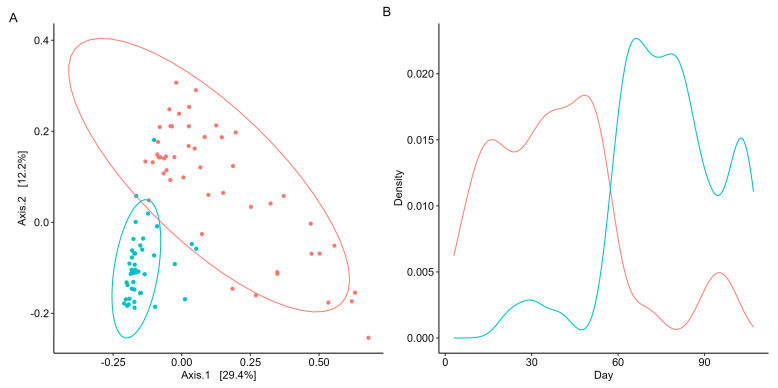
Unsupervised machine learning identifies two distinct microbial subcommunities. Cluster/enterotype 1 is in red, while cluster/enterotype 2 is represented by blue. (**A**) PCoA reveals the separation of samples based on microbial community structure and sampling days. Each point represents a sample day, with both axes explaining a specific percentage of variation in the data. The microbial community structure accounts for 29.4% of the variance, while the sampling day explains 12.2% of the distance between points. (**B**) Density plot visualizes the enterotypes’ distribution patterns to examine their prevalence and dominance throughout the participant’s swim.

**Figure 4 biomedicines-12-02309-f004:**
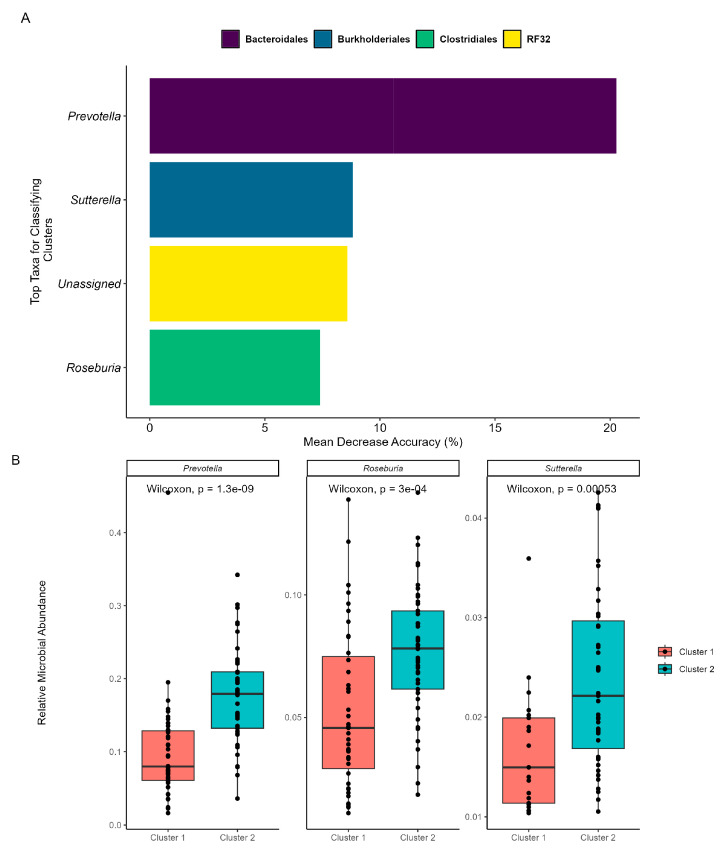
**Machine learning reveals important taxa within each enterotype.** The prediction model employed RF, was trained on the K-means categorizations, and was validated with OOB. (**A**) MDA of the RF highlights the top five taxa that exhibit shifts between the subcommunities. The taxa’s orders are categorized by color, and their genera are labeled on the *x*-axis. The “Unassigned” category includes bacteria not yet identified at the genus level. (**B**) Comparative analysis of relative abundance for the identified genera between cluster/enterotype 1 and cluster/enterotype 2.

**Table 1 biomedicines-12-02309-t001:** Subject physical characteristics and measures.

Variable	Quartile 1	Quartile 2	Quartile 3	Quartile 4	Counts	M ± SD	*p* Value
Daily swim distance	22.39 ± 11.78	20.24 ± 6.92	14.66 ± 6.86	12.58 ± 5.04	69	17.32 ± 8.44	**
Heart rate	64.4 ± 9.07	67.35 ± 5.43	64.09 ± 6.63	64.83 ± 8.33	39	65.67 ± 6.64	0.59
Blood pressure	115.4 ± 5.27	127.24 ± 9.71	133.45 ± 10.73	132.5 ± 14.18	39	128.28 ± 11.50	*
Body fat composition	21.00 ± 0.00	20.42 ± 1.01	18.19 ± 0.26	18.00 ± 0.00	27	19.31 ± 1.41	***

The participant’s daily swim distance, heart rate, blood pressure, and body fat composition were analyzed across the quartiles. Each quartile represents a segment of the study period, with the number of days as follows: Quartile 1 (27 days), Quartile 2 (27 days), Quartile 3 (26 days), and Quartile 4 (27 days). Summary statistics were calculated for each quartile, including observed counts, means, and standard deviations. An analysis of variance (ANOVA) was performed to compare the means between quartiles for each variable. The observed counts indicate the number of measurements available for each variable out of 94 data points. Significance levels are indicated: * *p* < 0.05, ** *p* < 0.01, *** *p* < 0.001.

## Data Availability

Data are publicly available here: https://github.com/marcell86/Machine-Learning-Reveals-Microbial-Taxa-Associated-with-a-Swim-Across-the-Pacific-Ocean (accessed on 29 May 2024).

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
