# Peer review of "Machine Learning Reveals Microbial Taxa Associated with a Swim across the Pacific Ocean"

_biomedicines, 2024, doi:10.3390/biomedicines12102309_

Round 1

Reviewer 1 Report

Comments and Suggestions for Authors

In this manuscript, the authors studied a person’s gut microbiome changes responding to excessive exercise (EE) by performing 16s rRNA analysis on the fecal samples. The authors observed no changes in alpha diversity and a drastic reduction in beta diversity. Specifically, using machine learning, the authors discovered a slight increase in Alistipes and decreases in Bifidobacterium and Ruminococcuss. A shift in subcommunities was observed from the start to 57 days and 58 days to the end. Increased presences of species like Pervotella, Roseburia, and Sutterella were found over the latter half of the exercise. The author also pointed out the limitations of the study such as the lack of dietary data, health data before the exercise period, BMR data, and unique exercise circumstances (a man swimming across the Pacific Ocean over 107 days). I think the manuscript is well written and provide valuable data and analysis for gut microbiome change over EE. With the minor concern addressed, I believe it is suitable for publication.

1.     The resolution for Figure 1 is low.

2.     Genus names in Figure 2 should be in italics. Also, the colors for Clostridiales and YS2 are too close, please change them to more contrasting colors.

3.     Genus names in Figure 4 should be in italics.

4.     Page 9. Reference #35 is in a different font or size to the rest of the text.

Author Response

Comments 1: The resolution for Figure 1 is low.

Response 1: Thank you for pointing this out. We agree with this comment. Therefore, we have changed the resolution of the figure to 300 dpi.

Comments 2: Genus names in Figure 2 should be in italics. Also, the colors for Clostridiales and YS2 are too close, please change them to more contrasting colors.

Response 2: Agree. We have, accordingly, changed the Genus names in Figure 2 to italics and changed the color palette so that Clostridiales and YS2 would be more distinct.

Comments 3: Genus names in Figure 4 should be in italics. 

Response 3: Agree. We have, accordingly changed the Genus names in Figure 4 to italics. 

Comments 4:   Page 9. Reference #35 is in a different font or size to the rest of the text.

Response 4: Font has been changed to match the format of the rest of the manuscript.

Reviewer 2 Report

Comments and Suggestions for Authors

In this study, Lewis et al. investigated the relationship between microbial dynamics and excessive exercise by analyzing fecal samples, body composition, and swimming logs from a single individual who swam across the Pacific Ocean over 107 days. Significant changes in microbial community structure and composition were revealed by the random forest machine learning method, highlighting specific microbes associated with increased exercise volume and its impact on body composition and blood pressure. This study presents a very interesting topic and the manuscript is well-written. There are only a few points that need further clarification. Here are some comments on this study:

1.        Section 2.2, after the fecal samples were obtained, how were they preserved before shipping, and were they in the preservation solution?

2.        It is necessary to indicate the name and version of the taxon annotation database.

3.        Table 1, “An analysis of variance (ANOVA) was performed to compare the means between quartiles for each variable”, I consider heart rate, blood pressure, and body fat composition data to be non-independent, and I would recommend using repeated measures ANOVA to perform the comparation.

4.        It is recommended that the author improve the resolution of the figures.

5.        Section 4.4, I quite agree that the lack of food-related data is a major limitation in this study because food has an important impact on gut microbiome. 

Author Response

Comments 1: Section 2.2, after the fecal samples were obtained, how were they preserved before shipping, and were they in the preservation solution?

Response 1: Thank you for pointing this out. We agree with this comment. Therefore, we have included this in methods: Fecal samples were stored in a -20 freezer during the swim. Following the conclusion, samples were shipped on dry ice to Argonne National Laboratory and stored in a -80 freezer until downstream processing. 

Comments 2: It is necessary to indicate the name and version of the taxon annotation database.

Response 1: We agree and have included the database version 13.8 within the methods section of the manuscript. 

Comments 3: Table 1, “An analysis of variance (ANOVA) was performed to compare the means between quartiles for each variable”, I consider heart rate, blood pressure, and body fat composition data to be non-independent, and I would recommend using repeated measures ANOVA to perform the comparation.

Response 3: We thank you for the comment. Heart rate, blood pressure, and body fat composition weren't collected with the same consistency as daily fecal samples. We employed an ANOVA to show overall variance in parameters associated with physical fitness, and our intention was to demonstrate a decline in exercise-associated output to demonstrate a correlation with changes in the gut microbiota.

Comments 4: It is recommended that the author improve the resolution of the figures.

Response 4: We agree. Therefore, the resolution of all figures has been changed to 300 DPI. 

Comments 5: Section 4.4, I quite agree that the lack of food-related data is a major limitation in this study because food has an important impact on gut microbiome. 

Response 5: We thank you for the comment and agree diet is a confounding variable.